# pH Sensing for Early Detection of Septic Inflammation Based on Intrinsic Titanium β-Alloy Nanotubular Oxides

**Jaroslav Fojt \*** , **Jan Šťovíček, Jitřenka Jírů** and **Vojtěch Hybášek**

Department of Metals and Corrosion Engineering, Faculty of Chemical Technology, University of Chemistry and Technology in Prague, Technická 5, 166 28 Prague, Czech Republic; stovicen@vscht.cz (J.Š.); jiruj@vscht.cz (J.J.); hybasekv@vscht.cz (V.H.)

**\*** Correspondence: fojtj@vscht.cz

**Abstract:** Orthopaedic surgeries like total hip and knee arthroplasty play a crucial role in restoring joint function for individuals with osteoarthritis. Deep bacterial infections are one of the most serious complications for orthopaedic implants. An infectious complication of an orthopaedic implant requires long-term and demanding treatment, often with repeated surgical procedures, and can lead to serious consequences such as implant failure, sepsis, and even death. Early detection of complications is of key importance for efficient therapy. The objective of this work is to investigate the possibilities of the nanotubular TiNbTa oxide for pH change sensing. Different surface treatments which lead to different surface natures were tested. For experiments, the inflammation was simulated by pH changes in the physiological solution. The response of the surface was monitored via the electrode potential changes. The results show that the nanotubular surface prepared on the TiNbTa alloy is a good possible candidate for pH sensing devices.

**Keywords:** pH sensing; nanotubes; titanium alloy; inflammation; electrode potential

## 1. Introduction

Orthopaedic surgeries such as total hip and total knee arthroplasty are vital procedures designed to restore joint function in individuals suffering from osteoarthritis. With the increasing life expectancy of the global population, the demand for these surgeries, which can significantly enhance quality of life, is also on the rise. Deep bacterial infections are one of the most serious complications of orthopaedic implants [1]. This complication results in pain, swelling, redness, and fistulation in the implant area. An infectious complication of an orthopaedic implant requires long-term and demanding treatment, often with repeated surgical procedures, and can lead to serious consequences such as implant failure, sepsis, and even death. Statistics indicate that prosthetic joint infections occur in approximately 1–2% of primary arthroplasty procedures and up to 4% of revision arthroplasty procedures [2].

In recent years, potentiometric sensors have shown remarkable promise in the field of analysis [3–5]. These sensors have exhibited outstanding analytical capabilities in detecting important targets. However, the current focus is on the development of in situ formats that enable the extraction of valuable information directly. The advantage of potentiometric assessment lies in its technical simplicity, affordability, and minimal space and energy requirements. Furthermore, potentiometric sensors are particularly well suited for the production of miniaturized sensors that offer satisfactory analytical performance. By leveraging these attributes, the aim is to enhance the practicality and accessibility of potentiometric sensor technology for various applications [6]. Detecting inflammation at an early stage is crucial for timely intervention and preventing further complications [7]. Implantable inflammation sensors provide real-time monitoring of inflammatory markers, allowing healthcare professionals to identify and address inflammation promptly. By detecting infections early, healthcare providers can initiate targeted antibiotic therapy or

even consider removing the implant if necessary, preventing the spread of infection and minimizing patient morbidity.

The most common microorganisms causing infections of orthopaedic implants are Gram-positive, facultatively anaerobic *Staphylococcus aureus* (golden staphylococcus) and *Staphylococcus epidermidis*; less common are Gram-negative aerobic *Escherichia coli*, *Proteus mirabilis*, and *Pseudomonas aeruginosa*. A serious problem nowadays is the increasing resistance of staphylococci to antibiotics [1]. One factor contributing to the predominance of staphylococci at the site of inflammation is their ability to switch to fermentative metabolism in situations of oxygen deprivation. Studies have shown that various inflammatory diseases are often associated with tissue hypoxia [8]. Most bacteria require iron to successfully infect human tissues, and iron is a key nutrient for them. Iron acquisition is difficult for bacteria because most iron in the body is tightly bound to proteins. Bacteria have therefore evolved a system that they activate during periods of starvation. Under these conditions, *S. aureus* redirects its metabolism using regulatory proteins, leading to the production of significant amounts of acidic products, especially lactate and formate. The accumulation of these acids facilitates the release of iron from the host proteins and at the same time causes a significant decrease in pH at the site of inflammation. As the local pH decreases, the oxidation–reduction potential is also reduced, leading to the conversion of insoluble host $Fe^{3+}$ to the more biologically advantageous $Fe^{2+}$ [9]. The change in pH could be used for early detection of infection and treatment, as the sensor would be part of the implant itself, for example, in hip replacements in the area of the extraction thread; however, we are aware of the potential problems associated with the miniaturization of the sensor.

When developing pH sensors, it is essential to consider several factors such as sensitivity, selectivity, pH measurement range, and durability. Metal oxides appear to be promising candidates for pH electrodes due to their fast response and long lifetime [6,10]. An interesting aspect is the possibility of using single metal oxides or modifying an existing alloy to obtain a mixture of oxides, thus improving the properties of the sensor. On the other hand, it should be mentioned that the way the oxides are prepared has a key influence on their behaviour [11,12]. By modifying the surface of the alloys, it is possible to obtain a much more stable structure and, under certain conditions, a real surface several times larger than the geometric one [12–14].

The main possible mechanisms of metal oxide ($MO_x$) sensitivity are:

- A redox equilibrium between two different solid phases;
- A redox equilibrium in one solid phase, which can be affected by the passage of current;
- A single-phase oxygen equilibrium;
- A simple ion exchange in a surface layer containing $OH^-$ groups.

Oxygen equilibrium provides the closest explanation for the mechanism of the sensitivity of $MO_x$ to pH [12]. Another proposed mechanism is that the pH response could be due to ion exchange in the surface layer. When the sensor is in contact with an aqueous environment, the $MO_x$ surface is covered with hydroxide groups from the dissociative adsorption of water. The release of protons can create oxide sites that lead to the formation of a pair of $MO_x$ molecules with a higher and lower valence and to a potential difference between the reference and working electrodes. The magnitude of this potential is proportional to the pH of the solution according to the Nernst equation, in this case, calculated at 37 °C (Formula (1)).

$$MO_x + 2\delta\,H^+ + 2\delta\,e^- \leftrightarrow MO_{x-\delta} + \delta\,H_2O$$

$$E = E^0 - 2.303\left(\frac{RT}{nF}\right)pH = E^0 - 0.062\,pH \tag{1}$$

$MO_x$ is the oxide in the higher oxidation state and $MO_{(x-\delta)}$ is the oxide with the lower oxidation state. E is the redox potential, $E^0$ is the standard electrode potential, R is the universal gas constant (8.314 J $K^{-1}$ $mol^{-1}$), T is the temperature 37 °C in K, F is the Faraday

constant (96,480 C mol$^{-1}$), and n is the number of electrons exchanged in the redox reaction. Thus, the theoretical sensitivity of the sensor to pH is given by 59 mV/pH [12,15].

Very promising candidates for modification are β-titanium alloys, for example, TiNbTa alloy. The metal oxides contained in this alloy are promising candidates. $Ta_2O_5$ has shown a Nernst response (59 mV/pH) and long-term stability in various works dealing with sensor fabrication issues [15–17]. However, these properties are highly dependent on the way the oxide layer is prepared [15]. In the work of M.J. Schöning and co-workers [18], they prepared a pH-sensitive sensor with $Ta_2O_5$ for online pH monitoring. The sensor prepared by them showed a sensitivity of 57 mV/pH in the pH range of 2–12, which was maintained over the long term.

The main element contained in the oxide layer of β-alloys is titanium. Titanium dioxide is an n-type semiconductor with a high chemical stability. At the same time, $TiO_2$ can be modified by anodization to obtain a nanotubular structure, which is characterized by a several times larger real area compared to the geometrical one, leading to a larger number of hydroxyl groups, which are involved in pH sensing [12,19,20]. In the work of Zhao R. et al. [19], $TiO_2$ nanotubes were created by anodic oxidation, and these tubes exhibited a near Nernstian response of 54 mV/pH and achieved a response of 59 mV/pH in the pH range of 2–12 after surface treatment with UV radiation.

Niobium oxides have not been greatly investigated in the available literature concerning their potential application in pH measurements [11,21]. In their work, Singewald et al. [11] tested three different electrode preparations with niobium oxides. Anodic polarization was found to be the most suitable preparation method, achieving a sensitivity of 41 mV/pH in the pH range of 2–12. The sensor prepared by them also showed excellent long-term stability.

From this information, it is concluded that by using Ti-36Nb-6Ta alloy to prepare a pH-sensitive surface, a stable and efficient pH sensor could be obtained.

## 2. Materials and Methods

Cylindrical Ti-36Nb-6Ta alloy samples (diameter: 16 mm, thickness: 3 mm) were wet ground (up to FEPA P2500 paper). The samples were sonicated in deionized water, isopropanol, and acetone and then dried in a stream of air. Anodic oxidation was carried out in an electrolyte containing 1 mol/L $H_3PO_4$ and 0.6 wt.% NaF. The experiments were carried out using a standard three-electrode setup with a platinum mesh as a counter electrode, the samples as a working electrode, and a silver/silver chloride reference electrode (3 mol/L KCl). All potentials mentioned in this article refer to this electrode. Nanostructuring consisted of a potential increase from the open circuit potential to a terminal potential of 20 V with a polarization rate of 10 mV/s, followed by holding the potential at the terminal potential for 2400 s, after potentiostatic exposure followed by potentiodynamic polarization to the open circuit potential (rate of 10 mV/s). Hydrothermal treatment was carried out in demineralized water at 100 °C for 1 h.

A scanning electron microscope (SEM) (MIRA2, Tescan, Brno, Czech Republic) was used for morphological characterization of the samples. The diameters and lengths of the nanotubes were determined by image analysis of three SEM image fields using FiJi 2.15.0 (ImageJ) software [22]. The surface composition of the samples was studied using an X-ray photoelectron spectrometer (XPS) ESCAprobe P (Omicron Nano-technology Ltd., London, UK) equipped with an Al Kα (E = 1486.7 eV) X-ray source. Spectra were measured with an energy step of 0.05 eV and normalized to the binding energy of the C1s peak (285.0 eV). CasaXPS software 2.3.15 (Casa Software Ltd., Devon, UK) was used to evaluate the spectra. Data for chemical state evaluation were obtained from the NIST X-ray photoelectron spectroscopy database [23]. The calculation of the oxide layer thickness is based on the free flight path of the electron between two inelastic collisions [24,25]. These calculated thicknesses are only approximate, only pure oxides were considered in the calculations, which, however, are not present in the examined samples; here, they are always mixtures.

Electrochemical measurements in response surface monitoring were performed on a Reference 600 potentiostat (Gamry, Warminster, PA, USA) in a standard three-electrode arrangement with a chloride-silver reference electrode and glassy carbon auxiliary electrodes. During surface stabilization testing, the open circuit potential (Eocp) and polarization resistance ($\pm$20 mV/Eocp) were recorded within a period of 24 h. After initial stabilization of the system (12 h) and before termination of exposure (120 h), the impedance spectrum (60 kHz-1 mHz, rms 10 mV/Eocp, 7 points per decade) was measured. The correctness of the impedance data was verified by a Kramers–Kronig transformation. Next, another overnight exposure was performed to estimate the Stern–Geary coefficient, ending with a linear polarization measurement ($-$50 mV/Eocp–1 V/SSCE). To measure the response of the surface to pH changes, the polarization resistance and open circuit potential were measured under the same conditions.

The model body environment was simulated with a physiological solution (9 g/L NaCl), the pH of which was adjusted using biological buffer TES (N-Tris (hydroxymethyl)methyl-2-aminoethanesulfonic acid, 5.9 g/L) and dilute solutions of NaOH and HCl. All electrochemical experiments were carried out at 37 °C.

## 3. Results

### 3.1. Description of Surface States

To test the possibility of pH sensing based on the intrinsic oxides of the support material, three variants of the oxide surface were prepared. The first simplest state was a native, self-forming layer on the surface of the material. Due to its natural origin, it can undergo significant fluctuations in properties, so for the second type of surface, it proceeded to stabilize its composition via a hydrothermal process. The third most complex surface is the formation of nanotubular oxide structures—so-called nanotubes. This surface has the advantages of increasing the sensor surface and a slightly different oxide structure.

The surface composition of the specimens determined via XPS is summarized in Table 1. Due to the bias of the results by the adsorbed contamination layer, oxygen and carbon were not included in the analysis. The results show that there is a depletion in the surface titanium in the passive layer. Here, the lower Gibbs free energy of the alloying metal oxides compared to $TiO_2$ is probably the cause [26]. In the case of a nanostructured surface, the difference is even more significant. Here, there is influence from the nanostructuring electrolyte used that contains fluoride ions. In such an environment, both alloying elements show a higher corrosion resistance than titanium [27], and hence, preferential etching occurs. An analysis of the detailed spectra showed that the native passive layer and the hydrothermally treated surface are composed of a mixture of oxides of each element (Figure 1) and their thickness does not exceed ten nanometres (due to the interaction depth of the XPS method), as the base metal signal is also detected. The measured spectra also show a restructuring of the passive layer. The signal of the sub-oxides of the individual metals was reduced on this modified surface and the oxides in the highest oxidation state show a clear dominance. In the case of the nanostructure, only the oxides of the elements at their highest oxidation state are present on the surface. The spectra were fitted with a maximum of the FWHM of 2 eV. This fitting procedure is in accordance with the standard approach [28,29]. The thickness of the layer was calculated using the Strohmeier and Carlson equations. However, the calculation of the thickness in this case is only indicative. Due to the impossibility of obtaining data for the free electron paths in such a complex system as the TiNbTa alloy and the resulting mixed oxide, the system was approximated in the calculations only to the presence of $TiO_2$ on the surface. However, even so, the increase in the thickness of the oxide layer is clearly evident not only from the calculations but also from the ratio of the peak intensities of the metals in the highest oxidation state and the zero oxidation state (Table 1).

The surface geometry characteristics of the nanostructured alloy are shown in Figure 2, where the histogram of the internal diameters is also shown. This parameter ranged from 15 to 60 nm, with the most frequent distribution between 30 and 40 nm. The length of the

nanotubes was $1690 \pm 72$ nm. These results are in agreement with previous work, where the mechanical properties and growth and differentiation of human mesenchymal cells, among other things, were tested [30].

**Table 1.** Surface composition determined via XPS.

| | Ti (%wt.) | Nb (%wt.) | Ta (%wt.) | F (%wt.) | Layer Thickness (nm) | Ti/TiO/Ti$_2$O$_3$/TiO$_2$ Atomic Ratio | Nb/NbO/NbO$_2$/Nb$_2$O$_5$ Atomic Ratio | Ta/Ta$_2$O$_5$ Atomic Ratio |
|---|---|---|---|---|---|---|---|---|
| nanotubes | 25 | 63 | 12 | | >10 | 100% TiO$_2$ | 100% Nb$_2$O$_5$ | 100% Ta$_2$O$_5$ |
| | 25 | 60 | 12 | 3 | | | | |
| hydrothermal treatment | 40 | 45 | 15 | | 9.7 | 2/5/5/88 | 0/7/10/83 | 50/50 |
| native layer | 44 | 50 | 6 | | 6.5 | 8/12/14/66 | 8/18/18/56 | 62/38 |
| bulk material | 58 | 36 | 6 | | | | | |

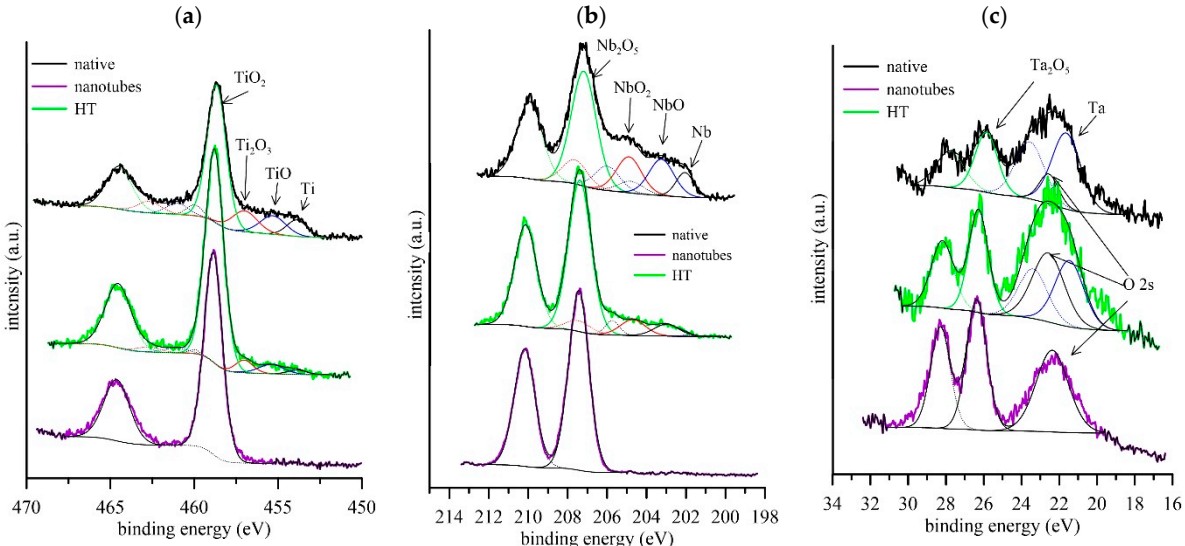

**Figure 1.** Comparison of the native and nanotube surface XPS spectra: (**a**) Ti 2p, (**b**) Nb 3d, (**c**) Ta 4f. The full lines are used for 2p3/2, 3d5/2, 4f7/2 and dashed lines for 2p1/2, 3d3/2, 4f5/2 peaks.

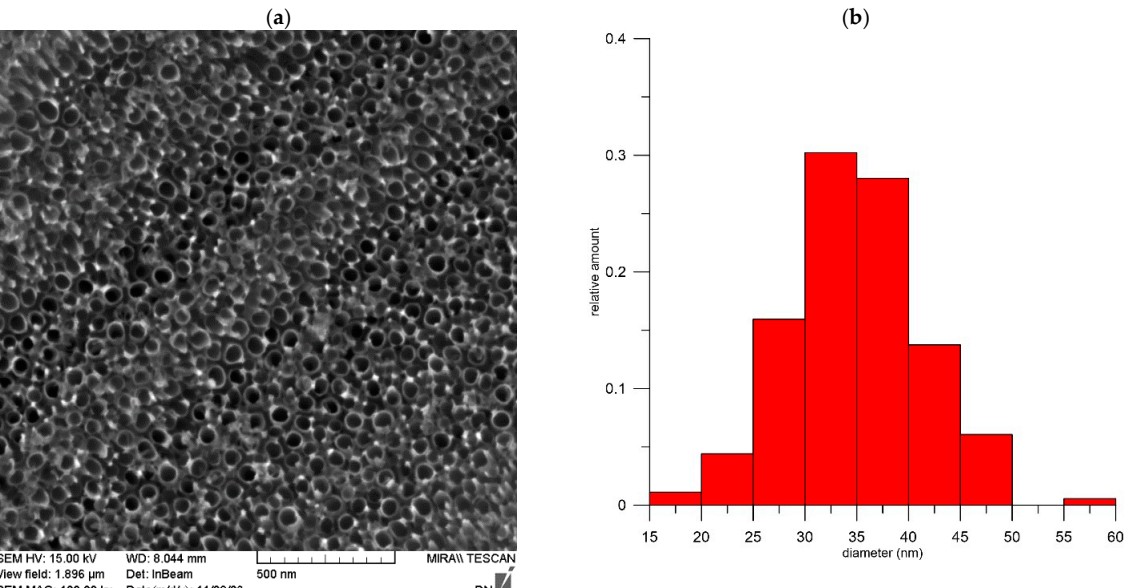

**Figure 2.** *Cont.*

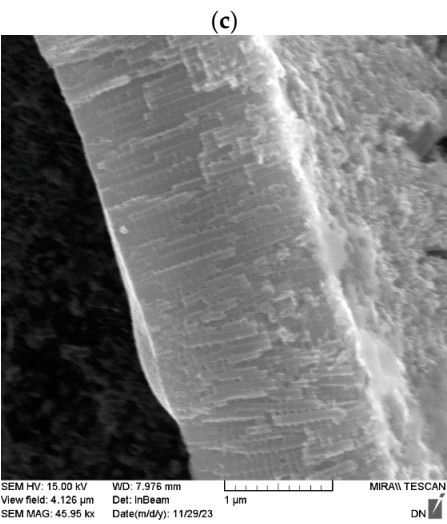

**Figure 2.** Nanotubular surface: (**a**) nanotubes top view, (**b**) histogram of the nanotube diameter distribution, (**c**) cross-section of the nanotubes.

### 3.2. Stability of Prepared Surfaces

A stable system over time in the environment is required to be able to measure the change in potential, reflecting the change in pH of the environment. For this reason, the studied surfaces were exposed to saline solution. The samples were monitored for 5 days regarding their open circuit corrosion potential and polarization resistance. The results of the measurements are summarized in Figure 3. In the case of the native surface, there was an increase in both the open circuit corrosion potential and the polarization resistance throughout the exposure. This is consistent with the results of electrochemical impedance spectroscopy. The impedance spectra were evaluated using an equivalent circuit that describes a compact oxide layer (Figure 4a,d) [31,32], where the resistance $R_1$ represents the charge transfer resistance and the $CPE_1$ (constant phase element) is equivalent to the combined capacitance of the passive layer and the electrical double layer. The goodness of fit ($\chi^2$), which is the calculated sum of the weighted residuals, was on the level of $10^{-4}$ or better. During the exposure, there was a threefold increase in the charge transfer resistance and a decrease in the layer capacitance (Table 2), i.e., an increase in the layer thickness and a slight decrease in the $\alpha$ parameter (an increase in the capacitance distribution), due to the increase in the layer non-uniformity. In the parallel measurement, the Stern–Geary coefficient was estimated to have a value of 0.043. In the case of the hydrothermally treated sample, there was a sharp increase in the open circuit potential during the first ten hours of exposure (Figure 3), with only a slight fluctuation thereafter. The polarization resistance increased slightly throughout the exposure, but the change was not as marked as in the case of the native surface. The impedance spectra of the hydrothermally treated sample confirm the minor evolution of the surface layer during the exposure. There was an increase in the charge transfer resistance; however, here, the fitting result is already affected by the significant error associated with the highly capacitive nature of the surface. For such surfaces, the impedance spectrum linearly increases in the low frequency region and no semicircle appears in the Nyquist presentation, so the fitting algorithm cannot reliably evaluate the charge transfer resistance. In such cases, its value is only indicative and changes of up to an order of magnitude are negligible. The capacitance value at the beginning of the exposure is lower than the capacitance of the native surface after 120 h (Table 2). It can therefore be assumed that the hydrothermal treatment has led to an increase in the thickness of the oxide layer, and it is possible to eliminate the need to stabilize the native layer in this way. There was a slight decrease in capacitance during exposure, indicating a restructuring of the surface, which is associated with an increase in the corresponding $\alpha$ value. The nanostructured surface showed stability from the beginning of the exposure, both in terms of open circuit potential and polarization resistance. Only minor fluctuations

were observed during the exposure without any obvious trend. The stability of the surface was also confirmed via impedance spectroscopy, with no significant changes in the spectra or the values of the individual elements used for evaluation during the exposure. The capacitance of the nanostructured surface was the lowest of all the treatments studied. It therefore corresponds to the thickest layer.

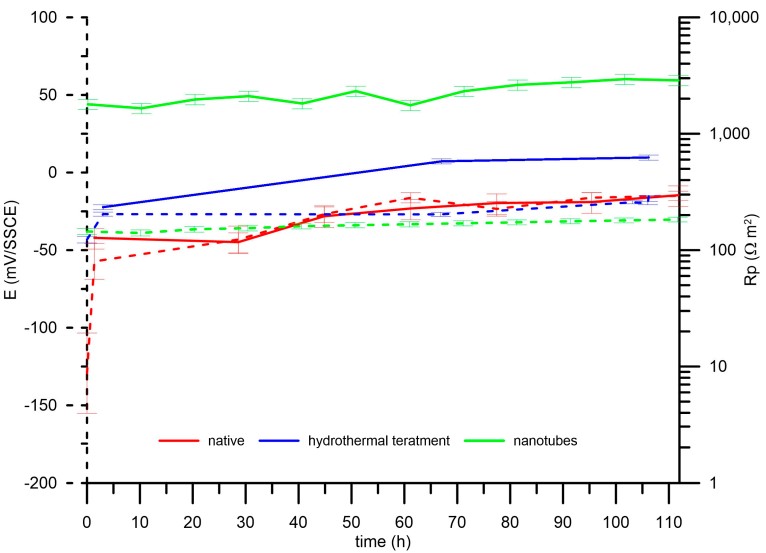

**Figure 3.** Initial surface stabilization in physiological solution, dashed lines: E, full lines: Rp.

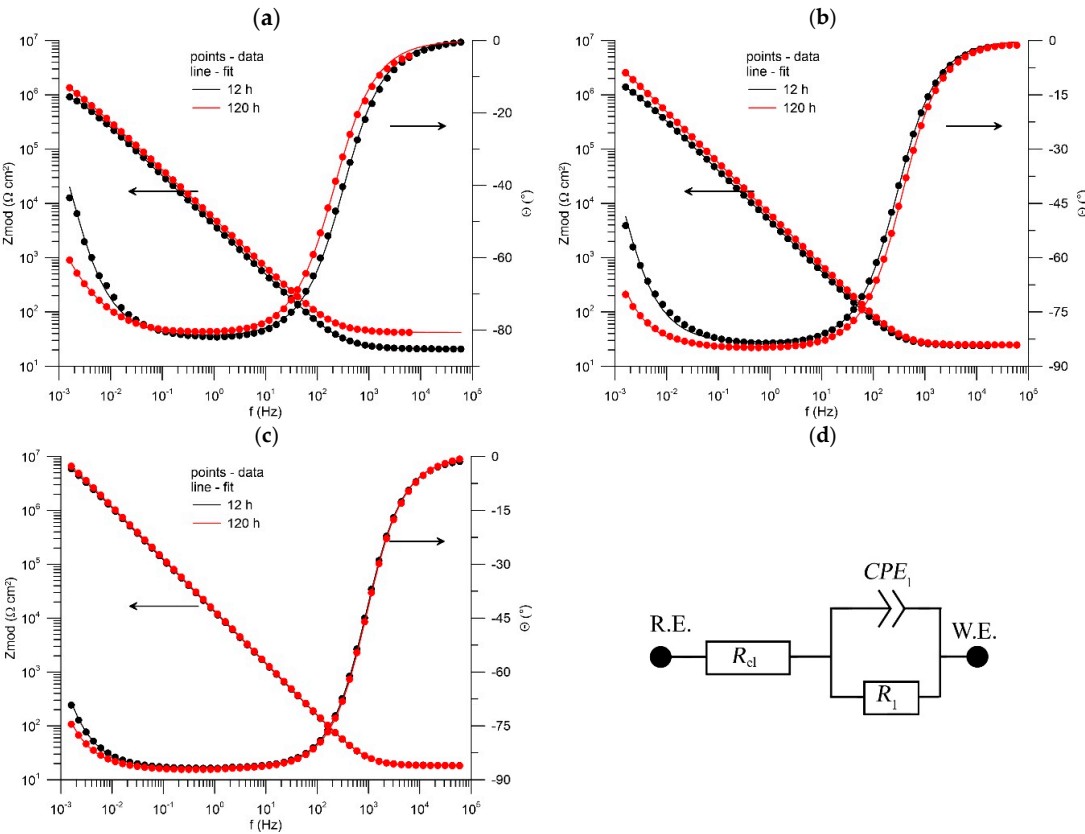

**Figure 4.** EIS spectra at the start and end of the stabilization period: (**a**) native passive layer, (**b**) hydrothermal treatment, (**c**) nanotubes, (**d**) equivalent circuit used for spectra analysis. The arrows indicate to which axis the curve is related.

**Table 2.** EIS evaluation.

| Surface | $R_1$ ($\Omega$ cm$^2$) | $CPE_1$ (S s$^\alpha$ cm$^{-2}$) | $\alpha$ | $C_{eff}$ (F cm$^{-2}$) | $\chi^2$ |
|---|---|---|---|---|---|
| native 12 h | $1.36 \times 10^6$ | $4.73 \times 10^{-5}$ | 0.910 | $7.14 \times 10^{-5}$ | $4.21 \times 10^{-4}$ |
| native 120 h | $3.53 \times 10^6$ | $3.71 \times 10^{-5}$ | 0.895 | $6.57 \times 10^{-5}$ | $4.42 \times 10^{-4}$ |
| HT 12 h | $2.41 \times 10^6$ | $3.87 \times 10^{-5}$ | 0.928 | $5.50 \times 10^{-5}$ | $2.32 \times 10^{-4}$ |
| HT 120 h | $9.46 \times 10^6$ | $2.78 \times 10^{-5}$ | 0.942 | $3.92 \times 10^{-5}$ | $9.64 \times 10^{-5}$ |
| nanotubes 12 h | $1.92 \times 10^7$ | $1.30 \times 10^{-5}$ | 0.964 | $1.60 \times 10^{-5}$ | $3.69 \times 10^{-5}$ |
| nanotubes 120 h | $2.97 \times 10^7$ | $1.24 \times 10^{-5}$ | 0.967 | $1.52 \times 10^{-5}$ | $3.07 \times 10^{-5}$ |

*3.3. Response to pH Changes*

Due to the instability of the native surface, it was not used further to test the sensing of pH changes. Thus, a hydrothermally treated surface that had undergone initial stabilization for 120 h in saline and a nanostructured surface without prior stabilization were tested. The pH levels were chosen based on the measurement of the pH of punctates from real patients with and without inflammation [2]. The time dependences of the open circuit potential and polarization resistance for the surfaces used are summarized in Figure 5. In the case of hydrothermal treatment, there was an increase in polarization resistance throughout the exposure time; however, this change did not exceed an order of magnitude. During the pH change, the surface responded with a change in potential, and this was at the level of 66 mV/pH, i.e., in the region of super Nernst behaviour. In the case of nanostructures, there was no significant change in polarization resistance; i.e., the system was stable. The potential changes with pH change reached the slope of 62 mV/pH; thus, Nernstian behaviour was achieved. In our previous work, data were published for a Ti-6Al-4V alloy, where a slope of only 32 mV/pH was achieved [2], and for a surface formed by combining a nanostructure with deposited iridium oxide particles, a slope of 64 mV/pH was achieved [33]. It is therefore evident that the newly investigated surface achieves equivalent results to much more complex systems. At the same time, the main advantage of using this procedure would be an overall reduction in the cost of producing such a sensor.

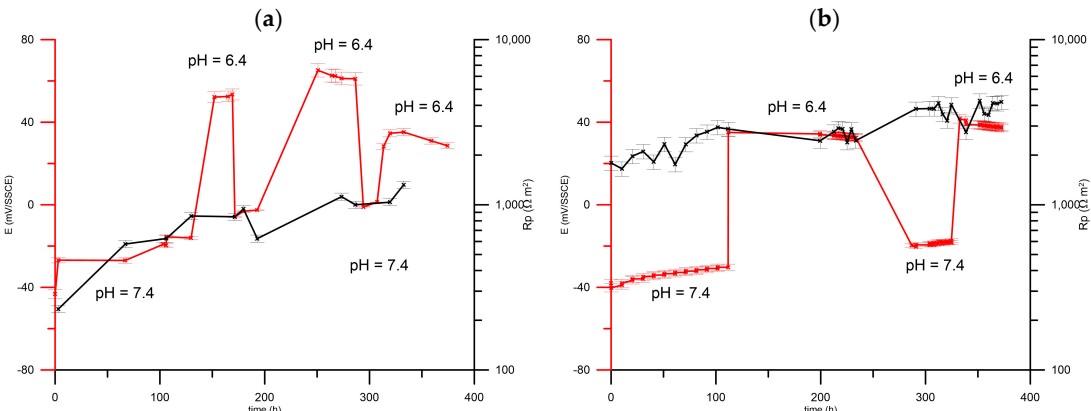

**Figure 5.** Time dependence of the electrode potential and polarization resistance during pH changes: (**a**) hydrothermal treatment, (**b**) nanotubes. Red line: E, black line: Rp.

Another essential parameter for pH sensors in the medical field is the rate of response to changing conditions. The rate was tested by exposing the sample to one pH electrolyte and then transferring it to a second pH electrolyte without rinsing the original medium. The results are summarized in Figure 6. In the case of a pH drop, both samples show a faster response; however, for the nanostructured sample, potential stabilization was achieved within ten minutes, while for the hydrothermal treatment, stability was reached after one hour. When the pH increased, both surfaces showed comparable behaviour, but also in

this case, stabilization occurred within an hour. In comparison with different systems, the reaction rate is slower [34,35]. However, given the intended application, the reaction rate is not critical and is compensated by long-term stability and ease of surface preparation.

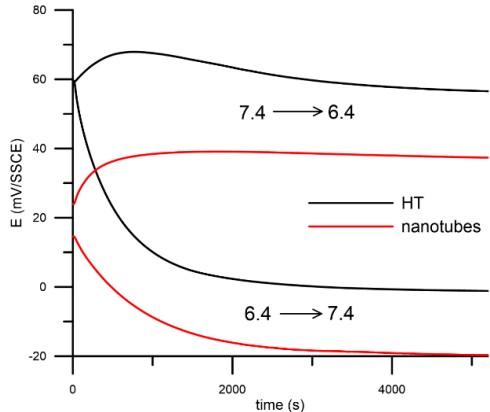

**Figure 6.** Rate of response to pH changes.

## 4. Discussion

The advantages of using proprietary oxides of chemically resistant, passive metals based on titanium alloys are considerable for implantable sensor materials. These are their non-toxicity, negligible risk of release (they are very cohesive), and extreme durability in the body environment. The chosen stabilization methods are well transferable from the laboratory to production.

The Ti-36Nb-6Ta alloy was chosen for two reasons. The first is the known properties and response of the cellular systems, even in the nanotopographically enhanced surface variant [36]. The second is the high content of niobium oxides in the layer, and niobium was chosen based on the possible structural change in the target pH range (Figure 7), which can be expected to affect the electrochemical response.

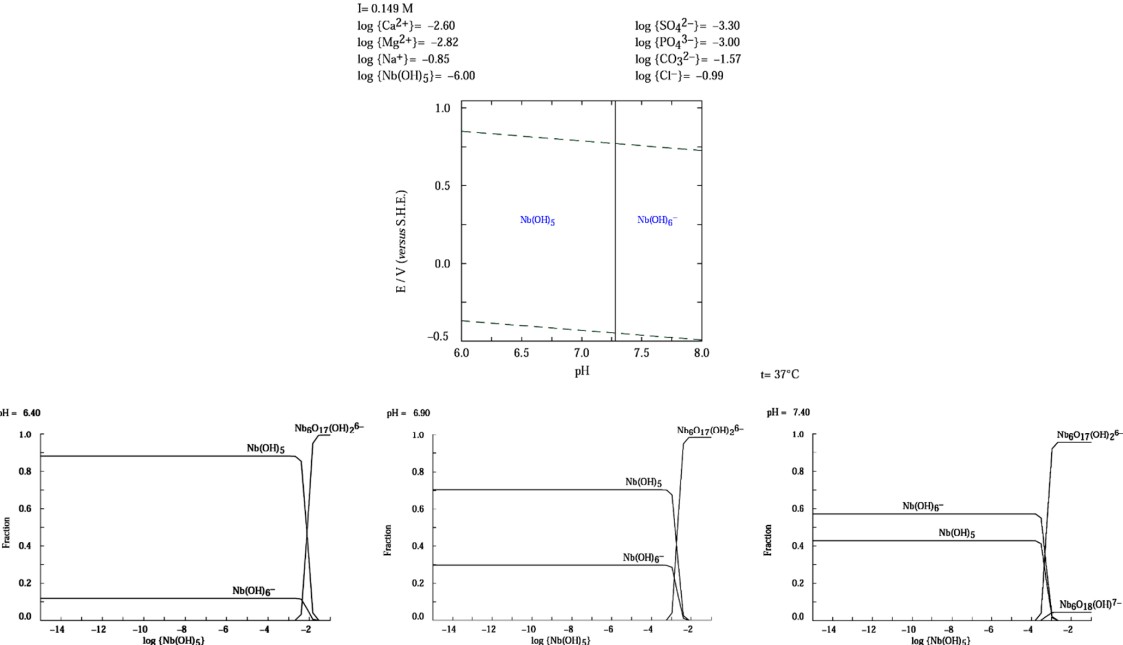

**Figure 7.** E–pH plot of niobium and fraction of its stable forms at different concentrations and pH in an inorganic blood plasma model, modelled in Hydra/Medusa [37]. The lines represent the equilibrium conditions for the compounds. The dashed lines represent the boundaries of the water stability region.

Due to the planned implantability and thus long-term function of the sensor, long-term stability is a priority; spontaneous transformations must not lead to a shift in the measured quantities that can be falsely interpreted as a change in the surrounding environment. Here, the influence of the native passive layer is evident, where, probably due to the high amount of metastable oxides ($Ti_2O_5$, $TiO$, $NbO_2$, $NbO$) in the initial state, significant structural changes occur, increasing the corrosion resistance (if the corrosion rate is calculated from the polarization resistance, this increase corresponds to a decrease in corrosion rate from $280 \text{ nm·a}^{-1}$ to $120 \text{ nm·a}^{-1}$). However, this leads to changes in all monitored parameters, which is why such a surface condition appears to be inappropriate. From the XPS spectra, it is evident that the hydrothermal treatment changes the ratio of suboxides and stable oxides within the formed layer in favour of $Nb_2O_5$ and $TiO_2$. However, even the treated alloy shows very similar changes in the electrochemical response (e.g., the corrosion rate decreased from $250 \text{ nm·a}^{-1}$ to $60 \text{ nm·a}^{-1}$), except for the open circuit potential, despite the higher initial layer thicknesses. Nonetheless, it is the measurement of the potential that is probably most easily realised under in-body conditions. A nanotubular surface, i.e., a surface containing cations in their stable state, shows only minimal changes with time.

The stability of the oxide layer also influenced the measurement of sensitivity to pH changes, which, for reasons of transferability to application conditions, was performed only using DC techniques. The significant change in polarization resistance for the hydrothermally treated material was also evident in the 14-day measurements with cyclic pH changes. There were clear fluctuations even for the previously stable OCP, with these changes resulting in significantly higher differences in the E/pH dependence from cycle to cycle.

Similarly to the stability measurements, the nanotube-based system appears to be inert in terms of polarization resistance (the corrosion rates of the sensor base material—assuming uniform dissolution through the oxide layer at the bottom of the nanotubes—are in the units of nanometers per year), which corresponds to the stability of sensitivity. A closer look at the rate of change in potential with a change in pH shows that the system begins to respond immediately with a full change in all cases within one hour, and realistically this value will be even lower, as there is an effect from the small charging current at the start of the measurement as well as the diffusive equilibration of the electrolyte pH.

Thus, from the aspects studied, the oxide nanotubes appear to be a sufficiently stable and pH-sensitive surface. In practical application, various interference effects such as biofouling may occur; however, it is possible to coat the sensor surface with a non-biofouling layer [2]. This study is taken as a pilot to identify theoretically applicable surfaces; these surfaces will then be tested on real punctates from patients with sterile and infected implants.

## 5. Conclusions

The native passive oxide layer contains metastable oxides such as $Ti_2O_5$, $TiO$, $NbO_2$, and $NbO$. During exposure in a physiological environment, changes in the passive layer occur, which are accompanied by a change in the corrosion potential. For this reason, it is not possible to use such a surface for the detection of pH changes.

Stabilization of the surface by a simple-to-implement hydrothermal treatment leads to an increase in oxide thickness and quality, and a sensitivity of 66 mV/pH is exhibited over the range of pH changes caused by septic inflammation. However, even such a stabilized surface is not inert.

The surface of the Ti-36Nb-6Ta alloy with nanotubes consisting mainly of $Nb_2O_5$, followed by $TiO_2$ and $Ta_2O_5$, showed long-term stability: there were no significant changes in the corrosion potential. This surface showed a sensitivity corresponding to a Nernst change of 62 mV/pH. The preparation of nanotubes on Ti-36Nb-6Ta alloy offers a simple option for sensing pH changes induced by septic inflammation.

**Author Contributions:** Conceptualization, J.F.; methodology, J.F. and V.H.; validation, J.F. and J.J.; formal analysis, J.Š., V.H. and J.F.; investigation, J.Š., V.H. and J.J.; writing—original draft preparation,

J.F., V.H. and J.J.; writing—review and editing, J.F., V.H. and J.J.; visualization, J.F.; supervision, J.F.; project administration, J.F.; funding acquisition, J.F. All authors have read and agreed to the published version of the manuscript.

**Funding:** This research was funded by the Czech Health Research Council, grant number NU20-06-00424.

**Data Availability Statement:** The data presented in this study are available on request from the corresponding author. The data are not publicly available due to privacy.

**Conflicts of Interest:** The authors declare no conflicts of interest.

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
