# Peer review of "pH Sensing for Early Detection of Septic Inflammation Based on Intrinsic Titanium β-Alloy Nanotubular Oxides"

_metals, doi:10.3390/met14020229_

Round 1
Reviewer 1 Report
Comments and Suggestions for Authors
The manuscript presents new original experimental results obtained by the authors in the process of studying, exploring the capabilities of TiNbTa nanotubular oxide for measuring pH changes.
Metal oxides are currently considered as promising candidates for pH electrodes. In this study, Ti-36Nb-6Ta alloy was used to prepare a pH-sensitive surface to create a stable and efficient pH sensor. Electrochemical measurements in surface of specimens were performed on a Reference 600 potentiostat (Gamry) in a standard three-electrode arrangement with a chloride-silver reference electrode and glassy carbon auxiliary electrodes.
The new results obtained expand knowledge about the effect of TiNbTa nanotubular oxide for measuring pH changes.
The title of the article correctly reflects the content of the study.
The main results formulated correspond to the data obtained.
The abstract correctly reflects the main ideas and results of the study.
The reference list is satisfactory. The reference list includes publications on the research direction in recent 5 years.
The results may be of interest to a wide range of specialists, graduate students and students studying the formation a nanotubular oxide on TiNbT alloy and it using for measuring pH changes.
The manuscript needs minor additions and clarifications of the presented results.
1) It is necessary to indicate the reliability intervals of the experimental data for the initial surface stabilization in physiological solution showed in Figure 3.
2) It is necessary to indicate the reliability intervals of the electrode potential and polarization resistance during pH changes showed in Figures 5 (a), (b).
3) The authors claim that «the fitting result is already affected by a significant error associated with the highly capacitive nature of the surface…». The impact of these errors on the data presented in Table 2 and on the conclusions of researches should be discussed in more detail.
4) The manuscript should be supplemented with an explanation of the parameter designations used in Table 2. Note that the parameter designations in Figure 4 and Table 2 have different interpretations (R1 and R1…, etc.).
The wording of the statements in the conclusion should be changed in order to specify the non-specific statements “it is not stable over the long term, it is not suitable as a pH sensor ...”, and also “The surface of Ti-36Nb-6Ta alloy with nanotubes consisting mainly of Nb2O5, followed by TiO2 and Ta2O5 already showed long-term stability...".
Reviewer 2 Report
Comments and Suggestions for Authors
The manuscript is interesting and reports the results of a study of the Ti-36Nb-6Ta alloy. Three different types of oxide layers were obtained on the alloy surface, and the reaction of the surface upon changing the pH of the environment was studied.
Some recommendations for improvement:
- I recommend adding SEM images of the native oxide layer and the oxide layer after hydrothermal treatment.
- The references used are not up-to-date. Only 4/32 references are from the last 3 years. If possible, add more references published in the last 3 years.
Author Response
- I recommend adding SEM images of the native oxide layer and the oxide layer after hydrothermal treatment.
Dear reviewer, the SEM images of the native and hydrothermally treated surface show no difference, due to the finish of the surface (ground on P2500). For this reason, we decided not to include the images in the manuscript.
- The references used are not up-to-date. Only 4/32 references are from the last 3 years. If possible, add more references published in the last 3 years.
The references were updated, now, there is 32% of the references not older than 2021.
Reviewer 3 Report
Comments and Suggestions for Authors
I have carefully reviewed your manuscript and found several points that I believe require further attention and clarification. I kindly request that you consider the following major review points for the improvement of your manuscript:
1. The concept of how the sensor, as part of the implant, would effectively indicate low pH levels is not clearly explained in the manuscript. Please provide a more detailed explanation of how the sensor would function within the implant environment to detect pH changes.
2. The XPS spectra presented in Figure 1 are unclear, as the peaks appear to contain overlapped contributions and exhibit different FWHM values. I recommend revisiting the XPS analysis and providing clearer and more interpretable spectra.
3. There is no disclosure of ethical committee approval for the use of data from real patients, as mentioned in line 242 of the manuscript. Please provide the necessary information to support the ethical use of patient data in your study.
4. The in vitro clinical situation is significantly more complex than the simulation described in your manuscript. Therefore, it is important to discuss the limitations of your data in accurately representing the in vivo clinical environment.
5. The majority of the references cited in the manuscript are dated and do not reflect the ongoing literature in this field. I suggest updating the references to include more recent and relevant sources.
I believe that addressing these points will significantly enhance the quality and clarity of your manuscript.
